A close relative of the Amazon river dolphin in marine deposits: a new Iniidae from the late Miocene of Angola

Lambert Olivier olivier.lambert@naturalsciences.be 1
Auclair Camille 2 3
Cauxeiro Cirilo 4 5
Lopez Michel 5
Adnet Sylvain 2
1 D.O. Terre et Histoire de la Vie, Institut royal des Sciences naturelles de Belgique , Brussels , Belgium
2 ISEM, Université de Montpellier , Montpellier , France
3 Sardent , France
4 Faculdade de Ciências, Universidade Agostinho Neto , Luanda , Angola
5 Geosciences, Université de Montpellier , Montpellier , France
Hrbek Tomas
Electronic publication date: 2018 Sep 12
Publication date: 2018
Volume: 6
Electronic Location ID: e5556
Received 2018 Mar 16; Accepted 2018 Aug 9
Copyright: ©2018 Lambert et al.
Copyright year: 2018
Copyright holder: Lambert et al.
License: This is an open access article distributed under the terms of the Creative Commons Attribution License, which permits unrestricted use, distribution, reproduction and adaptation in any medium and for any purpose provided that it is properly attributed. For attribution, the original author(s), title, publication source (PeerJ) and either DOI or URL of the article must be cited.
License URL: https://creativecommons.org/licenses/by/4.0/

Keywords: Cetacea, Inioidea, Iniidae, Angola, late Miocene, Kwanza basin, Amazon river dolphin

Funding: TOTAL Exploration Production Angola (TEPA) The discovery and excavation of the fossil material studied in this work was funded by TOTAL Exploration Production Angola (TEPA). The funders had no role in study design, data collection and analysis, decision to publish, or preparation of the manuscript.

==============================
Background

A few odontocetes (echolocating toothed cetaceans) have been able to independently colonize freshwater ecosystems. Although some extant species of delphinids (true dolphins) and phocoenids (porpoises) at least occasionally migrate upstream of large river systems, they have close relatives in fully marine regions. This contrasts with the three odontocete families only containing extant species with a strictly freshwater habitat (Iniidae in South America, the recently extinct Lipotidae in China, and Platanistidae in southeast Asia). Among those, the fossil record of Iniidae includes taxa from freshwater deposits of South America, partly overlapping geographically with the extant Amazon river dolphin Inia geoffrensis, whereas a few marine species from the Americas were only tentatively referred to the family, leaving the transition from a marine to freshwater environment poorly understood.

Methods

Based on a partial odontocete skeleton including the cranium, discovered in late Miocene (Tortonian-Messinian) marine deposits near the estuary of the Cuanza River, Angola, we describe a new large iniid genus and species. The new taxon is compared to other extinct and extant iniids, and its phylogenetic relationships with the latter are investigated through cladistic analysis.

Results and Discussion

The new genus and species Kwanzacetus khoisani shares a series of morphological features with Inia geoffrensis, including the combination of a frontal boss with nasals being lower on the anterior wall of the vertex, the laterally directed postorbital process of the frontal, the anteroposterior thickening of the nuchal crest, and robust teeth with wrinkled enamel. As confirmed (although with a low support) with the phylogenetic analysis, this makes the new taxon the closest relative of I. geoffrensis found in marine deposits. The geographic provenance of K. khoisani, on the eastern coast of South Atlantic, suggests that the transition from the marine environment to a freshwater, Amazonian habitat may have occurred on the Atlantic side of South America. This new record further increases the inioid diversity during the late Miocene, a time interval confirmed here as the heyday for this superfamily. Finally, this first description of a Neogene cetacean from inland deposits of western sub-Saharan Africa reveals the potential of this large coastal area for deciphering key steps of the evolutionary history of modern cetaceans in the South Atlantic.

Introduction

Many recent works based on both morphological and molecular arguments have demonstrated and elaborated on the iterative, independent shift of echolocating toothed cetaceans (odontocetes) from marine to freshwater environments (e.g., Fordyce, 1983; Muizon, 1988; Cassens et al., 2000; Nikaido et al., 2001; Geisler et al., 2011; Geisler, Godfrey & Lambert, 2012; Bianucci et al., 2013; Gutstein, Cozzuol & Pyenson, 2014; Gutstein et al., 2014; Pyenson et al., 2015; Aguirre-Fernández et al., 2017).

The broad distribution and genetic diversity of the Amazon river dolphin (Inia spp., family Iniidae) in South American freshwater ecosystems suggests a long evolutionary history in this vast region (Best & Da Silva, 1989; Hrbek et al., 2014). Although several species of Inioidea (superfamily including Iniidae + Pontoporiidae) from late Miocene and early Pliocene marine deposits of South and North America were tentatively referred to the Iniidae (Allen, 1941; Muizon, 1988; Cozzuol, 2010; Geisler, Godfrey & Lambert, 2012; Pyenson et al., 2015; Lambert et al., 2017), the content of the family is still debated, either due to the fragmentary state of the type material or to the lack of clear diagnostic features. The only extinct iniid displaying the typical vertex of the cranium of the extant species is Ischyrorhynchus vanbenedeni Ameghino, 1891, originating from freshwater late Miocene deposits of Argentina and Brazil (Cozzuol, 2010; Gutstein, Cozzuol & Pyenson, 2014); the lack of iniid fossils bearing such a typical vertex in marine deposits leaves one of the key steps of the family’s evolutionary history, the marine to freshwater transition, poorly resolved.

In this work we report on the description of a new iniid, based on a partial skeleton including the cranium of a relatively large dolphin from late Miocene marine deposits of Angola. It originates thus from a region, sub-Saharan Africa, that up to now only produced a very limited number of cetacean fossils (e.g., Andrews, 1919; Bianucci, Lambert & Post, 2007; Jacobs et al., 2016; Mourlam & Orliac, 2018; see also a detailed review in Gingerich, 2010, including a couple of records from Angola). Displaying a structure similar to the frontal boss typical of the extant Inia, the new genus and species brings elements feeding the debate on the phylogenetic relationships and palaeobiogeographic history of Iniidae.

Materials and Methods

Nomenclatural acts

The electronic version of this article in portable document format (PDF) will represent a published work according to the International Commission on Zoological Nomenclature (ICZN), and hence the new names contained in the electronic version are effectively published under that code from the electronic edition alone. This published work and the nomenclatural acts it contains have been registered in ZooBank, the online registration system for the ICZN. The ZooBank LSIDs (life science identifiers) can be resolved and the associated information viewed through any standard web browser by appending the LSID to the prefix http://zoobank.org/. The LSID for this publication is: urn:lsid: zoobank.org:pub:9488E279-A53A-4E7A-A2F7-AF8C693C208A. The online version of this work is archived and available from the following digital repositories: PeerJ, PubMed Central and CLOCKSS.

Studied specimen

The specimen consists of a partly preserved cranium (CZA 1), a detached fragment of the rostrum including both premaxillae, the right maxilla, and two teeth (CZA 2), three posterior teeth on a smaller right maxilla fragment (CZA 3), the axis (CZA 4), a more posterior cervical vertebra (CZA 5), a vertebral centrum (CZA 6), fragments of forelimb (CZA 7 and CZA 8), and fragments of ribs (CZA 9 to 16). Considering the fragmentary state of other elements, not preserving any diagnostic feature, only the cranium, teeth, and cervical vertebrae are described here. Material is temporarily housed at ISEM for study, before its final return to the Universidade Agostinho Neto, Luanda, Angola.

Other inioid and lipotid specimens directly observed

Brachydelphis mazeasi Muizon, 1988: MNHN PPI 121, 124, 266, MUSM 564, 589, 591, 593, 886, 887, 2473, 2539; B. jahuayensis Lambert & Muizon, 2013: MNHN PPI 267, 268, MUSM 567, 568, 712, 884, 2611 (=AGL 141), MUSM 2618 (=AGL 732); Brujadelphis ankylorostris Lambert et al., 2017: MUSM 1400; Inia geoffrensis (Blainville, 1817): USNM 395614, ZMA 17.771, several unnumbered specimens at MNHN (4), MUSM DPV CE 9, and an additional unnumbered specimen at MUSM; Lipotes vexillifer Miller, 1918: USNM 218293; Meherrinia isoni Geisler, Lambert & Godfrey, 2012: CMM-V-4051, 4052, 4060, IRSNB M.2013; Pliopontos littoralis Muizon, 1983: MNHN SAS 193, 931, 953, MUSM 953, 976, 6253; cf. Pontistes sp.: cast of MGUH 1922-168; Pontoporia blainvillei (Gervais & d’Orbigny, 1844): IRSNB 1506, ZMA 16.714, USNM 482772; Protophocaena minima Abel, 1905: IRSNB 3917-M.172, M.2303, NMB 1, TM 25112; Stenasodelphis russellae Godfrey & Barnes, 2008: CMM-V-2234.

In addition, remains of multiple other early delphinidans from the Miocene have been directly observed, especially at the CMM, IRSNB, MNHN, and USNM. Finally, 3D models of crania of extant odontocetes, including I. geoffrensis (USNM 239667, 395415, and 49582) and P. blainvillei (USNM 482727, 482771, 482763), were accessed via phenome10k.org, and 3D models of the holotype of Isthminia panamensis Pyenson, Vélez-Juarbe, Gutstein, Little, Vigil & O’Dea, 2015 via 3d.si.edu.

Anatomical terminology

Terminology for cranial anatomy follows Mead & Fordyce (2009) unless otherwise stated.

Phylogenetic analysis

To investigate the phylogenetic relationships of the new taxon we coded CZA 1-16 and the recently described late Miocene inioid Isthminia panamensis in the character/taxon matrix of Post, Louwye & Lambert (2017), modified from the matrix of 324 characters by Lambert et al. (2017) (see Supplemental Information). Ordering of part of the characters was done following the criteria detailed with a previous version of this matrix (Geisler et al., 2011). As in Post, Louwye & Lambert (2017), using PAUP 4.0a (Swofford, 2003) three outgroups (Bos taurus, Hippopotamus amphibius, and Sus scrofa) were defined; ordered multistate characters were scaled for a minimum length of each being one step; and a constraint tree resulting from Bayesian analysis of a molecular dataset on extant species was enforced as a backbone (see Supplemental Information). Most parsimonious trees were obtained by heuristic search, using tree-bisection-reconnection branch swapping algorithm and ACCTRAN character-state optimization.

Geological Setting

The fossil material studied here was discovered south of the estuary of the Cuanza River (Barra do Cuanza), in the Inner Kwanza basin, 74 km south of Luanda (GPS Bc65: 9°21′33,47″S 13°9′10,50″E; Altitude 1,00 m) (Fig. 1). The Inner Kwanza basin corresponds to a salt-controlled mobile margin submitted to an overall uplift during the late Neogene that led to the continentalization of the domain and the outcropping of shelf to slope deposits along well-exposed coastal cliffs. In particular, south of Barra do Cuanza the lower to middle Miocene deposits were slightly tilted to the south during the raft tectonic and were overlapped unconformably by upper Miocene-Pliocene sands, where the skeleton was discovered (Fig. 2). The lower to middle Miocene (Fig. 2: sequence 1) of the tilted block was dated by a foraminiferal and calcareous nannoplankton assemblage (Cauxeiro, 2013). It shows a partly preserved regressive sequence passing from dark to black organic-rich shales alternating with poorly graded fine to medium hummocky cross-stratified sandstone from offshore transition environment, to clayey mudstone-blue-grey marls alternations of more protected outer shelf environment on top. This series is truncated by a dark orange to brown iron-rich 50 cm-thick fossil-rich lag conglomerate reworking the previous deposits and indicating a major time gap and associate condensation episode (Fig. 2: sequence 2). This regional marine erosional surface was dated from the middle Miocene (late Langhian) by Cauxeiro (2013). It is unconformably overlapped by an overall transgressive sequence showing in the lower part 30 to 40 m thick well sorted, fine to medium beige sand, partly indurated, and intensely burrowed by Ophiomorpha (Fig. 2: sequence 3). These deposits are intersected by 2D megaripple bodies of medium to coarse sand indicating an overall northward transport by longshore currents in the upper shoreface domain. The upper part of the sequence (Fig. 2: sequence 4) is mainly composed of about 20 m of greenish grey, intensely bioturbated clayey silt to fine sands that indicate the deepening of the depositional system towards the lower shoreface domain. The odontocete remains were discovered in the lower sand unit (Fig. 2: sequence 3), about 10 m above the basal erosional unconformity. The skeleton elements were likely partly connected or in situ dismembered as demonstrated by the discovery of successive vertebrae behind the skull (Fig. 3). The upper shoreface sands containing the fossil could not be dated directly, their dating thus remains relatively uncertain inside the Tortonian-Messinian interval (Cauxeiro, 2013). The top of the sedimentary pile of the cliff shows a 20 to 30 m thick coarsening upward sequence of poorly sorted fine to coarse sand (Fig. 2 sequence 5). The base of this sequence is sharp and corresponds to a lag conglomerate of gravels and reworked bioclasts and cetacean bones. This last level probably corresponds to the fossiliferous levels described by Jacobs et al. (2016) 5 km north of the Cuanza River mouth, with the presence of fossil mysticete and crocodile remains. This sequence is reported to the progradation of the proto-Cuanza deltaic prism that follows the major late Miocene sea level fall during the early Pliocene highstand (Cauxeiro, Durand & Lopez, 2014).

Figure 1 Locality of the holotype of Kwanzacetuskhoisani.

Location map and geological overview of the discovery zone of K. khoisani in the Kwanza basin, Angola, modified from Cauxeiro, Durand & Lopez (2014). Image credit: Sylvain Adnet and Michel Lopez.

Figure 2 Stratigraphical context for the holotype of Kwanzacetus khoisani.

Stratigraphical architecture of the cliff south of Barra do Cuanza where the holotype of K. khoisani was discovered. (A) panoramic view; (B) interpreted line drawing; and (C) sedimentary column showing the main facies assemblage. In the lower part of the cliff, alternating hummocky cross stratified sandstones, mudstones, and black shales from the lower to middle Miocene (sequence 1) are slightly tilted southwards and obliquely truncated by the major erosional unconformity (sequence 2). This erosional surface is onlapped by an overall fining upward sequence dated from the late Miocene. This sequence shows upper shoreface sand burrowed by Ophiomorpha (sequence 3) passing upwards to burrowed fine sand to clayey silt from lower shoreface environment (sequence 4). The upper part of the cliff is composed of an upward coarsening sequence, from fine to coarse sand and gravel (sequence 5), which marks the overall progradation of the paleo-Cuanza delta during the Pliocene. This sequence is floored by a marine ravinement lag deposit. Abbreviations: F fossil locality (see also Fig. 3); M mudstone; W wackestone; P packstone; G grainstone; B boundstone; C clay; Si silt; f fine sand; m medium sand; c coarse sand; G gravel; HCS hummocky cross stratification; TaCS tabular cross stratification; TrCS trough cross stratification. Image credit: Sylvain Adnet and Michel Lopez.

Figure 3 Location of the holotype of Kwanzacetuskhoisani on the cliff (indicated in Fig. 2).

Photographs before (A) and after (B) the extraction of the skull. Orange Fe-hydroxide halos in the sand indicate Ophiomorpha burrowing. Photo credit: Michel Lopez.

Fossil material collected by some of the authors (ML, CC) during a field trip in 2012 was prepared in the laboratory (CA), removing its indurated sandy matrix with an airscribe. It consists of a partial skull with connected fragment of rostrum, some vertebrae, and fragments of ribs and forelimbs.

Systematic Paleontology

Cetacea Brisson, 1762	
Pelagiceti Uhen, 2008	
Neoceti Fordyce & Muizon, 2001	
Odontoceti Flower, 1867	
Delphinida Muizon, 1984	
Inioidea Muizon, 1988	
Iniidae Gray, 1846	
Kwanzacetus, gen. nov.	

Type and only included species: Kwanzacetus khoisani, sp. nov.

Etymology: Kwanza from the Kwanza basin, the area where the holotype was collected; cetus, whale in Latin.

Diagnosis: Same as for the only included species.

Kwanzacetus khoisani, sp. nov.	

Holotype and only referred specimen: CZA 1-16, a partial skeleton including the partly preserved cranium, five teeth, the axis, a more posterior cervical vertebra, a vertebral centrum, fragments of forelimb and ribs.

Type locality: Kwanza basin - South Barra do Cuanza and estuary of the Cuanza River (Figs. 1– 2), about 74 km south of Luanda (GPS Bc65: 9°21′33, 47″S 13°9′10, 50″E), Angola.

Type horizon: The holotype was discovered about 10 m above the basal erosional unconformity of an unnamed lithological unit, made of fine to medium-grained beige sand, partly indurated and intensely burrowed by Ophiomorpha, which constitutes the lower part of a local transgressive sequence (Fig. 2). A Tortonian-Messinian (late Miocene) age interval can be proposed for the level where the specimen was discovered (see details above).

Etymology: From Khoisan, an indigenous people formerly inhabiting the Angolan coast.

Diagnosis of species: This large Iniidae (bizygomatic width of the holotype estimated at 286 mm) shares with the smaller Brujadelphis ankylorostris Lambert et al., 2017 and Isthminia panamensis: the partial ankylosis of the premaxillae on the rostrum; with Inia geoffrensis and Ischyrorhynchus vanbenedeni: the presence of a frontal boss, with nasals being lower than the frontals on the vertex; and teeth being markedly ornamented, with wrinkled enamel; with I. geoffrensis: the laterally directed postorbital process of the frontal; the anteroposterior thickening of the nuchal crest (to an even greater extent than I. geoffrensis); and the more developed left occipital protuberance (the latter being only seen in part of the I. geoffrensis sample, including specimens directly observed, 3D models, and illustrations/descriptions in the literature, see below).

It further differs from Isch. vanbenedeni in: the nasals being anteroposteriorly longer, broadly exposed dorsally; the frontal boss being considerably lower, only a few millimeters higher than the nuchal crest; and the postglenoid process being located higher than the paroccipital process. It further differs from I. geoffrensis in: the lower premaxillary eminence, lacking a vertical lateral wall; the postorbital process being anteroposteriorly and transversely thick; the nasals being anteroposteriorly longer, broadly exposed dorsally; the frontal boss being considerably lower, only a few millimeters higher than the nuchal crest; the temporal crests projecting far posterior to the supraoccipital shield; and the absence of any heel on the crown of posterior maxillary teeth.

Among inioid species at least tentatively attributed to the family Iniidae, it further differs from Brujadelphis ankylorostris and Isthminia panamensis in: a broadly dorsally exposed squamosal fossa; and the temporal crests projecting far posterior to the supraoccipital shield; from Meherrinia isoni in: the presphenoid being barely exposed between the premaxillae; nasals being approximately as wide as the bony nares; the frontals on the vertex being at least as long as transversely wide; and the absence of an anterodorsomedial projection of the supraoccipital shield between the supraorbital processes; from the poorly preserved Goniodelphis hudsoni Allen, 1941 in: left and right upper alveolar groove remaining distant from each other anteriorly; the absence of a markedly concave lateral margin of the premaxilla in the antorbital area; and the maxilla being roughly as wide as the premaxilla at the level of the antorbital notch

Description and Comparison

Cranium

Preservation state

The proximal part of the rostrum is only partly preserved, with most of the lateral margins incomplete; both antorbital notches are lost, as well as most of the left supraorbital region, the right preorbital region, the left lateral part of the basicranium, parts of the right squamosal, ear bones, and the most fragile bones of the ventral surface (Figs. 4– 9). A part of the neural arch of a vertebra is attached in the upper part of the right temporal fossa and a smaller fragment of bone partly covers the dorsomedial region of the supraoccipital shield.

Figure 4 Dorsal view of the cranium of Kwanzacetus khoisani.

Photograph (A) and corresponding line drawing (B) of the cranium of the holotype of K. khoisani CZA 1-2 in dorsal view. Light grey for sediment and plaster; dark grey for attached bone fragments; hatched areas for major break surfaces; dotted lines for interpretation of unclear sutures. Scale bar equals 100 mm. Photo credit: Olivier Lambert.

Figure 5 Lateral view of the cranium of Kwanzacetus khoisani.

Photograph (A) and corresponding line drawing (B) of the cranium of the holotype of K. khoisani CZA 1-2 in right lateral view. Light grey for sediment and plaster; dark grey for attached bone fragments; hatched areas for major break surfaces; dotted lines for interpretation of unclear sutures. Scale bar equals 100 mm. Photo credit: Olivier Lambert.

Figure 6 Ventral view of the cranium of Kwanzacetus khoisani.

Photograph (A) and corresponding line drawing (B) of the cranium of the holotype of K. khoisani CZA 1 in ventral view. Note that the detached, tooth-bearing anterior part of the rostrum figured in dorsal and right lateral view is not present here. Light grey for sediment and plaster; hatched areas for major break surfaces. Scale bar equals 100 mm. Photo credit: Olivier Lambert.

Figure 7 Posterior view of the cranium of Kwanzacetus khoisani.

Photograph (A) and corresponding line drawing (B) of the cranium of the holotype of K. khoisani CZA 1 in ventral view. Light grey for sediment and plaster; dark grey for attached bone fragments; hatched areas for major break surfaces. Scale bar equals 100 mm. Photo credit: Olivier Lambert.

Figure 8 Detail of the basicranium, orbit, and palate of the cranium of Kwanzacetus khoisani.

Photograph (A) and corresponding line drawing (B) of the right side of the basicranium, orbit, and palate of the holotype of K. khoisani CZA 1-2 in ventrolateral and slightly anterior view. Light grey for sediment and plaster; dark grey for attached bone fragments; hatched areas for major break surfaces. Scale bar equals 100 mm. Photo credit: Olivier Lambert.

Figure 9 Additional views of the cranium of Kwanzacetus khoisani.

(A) right posterolateral and slightly dorsal view of the holotype of K. khoisani CZA 1-2; (B) anterior and slightly dorsal view; (C) right anterolateral and slightly dorsal view. Dotted lines for main sutures and other bone outlines. Scale bar equals 100 mm. Photo credit: Olivier Lambert.

Surfaces of the bones are often considerably abraded and a thin layer of sediment is retained in a few regions, due to the low mineralization of underlying bone, making the latter extremely delicate. Low mineralization degree presumably lead to the formation of a few cavities, the largest being located on the right maxilla lateral to the vertex. As a consequence of this preservation state, part of the sutures and at least some of the cranial foramina could not be detected.

Taking account of the moderate deformation of the foramen magnum and the left temporal crest being closer to the corresponding occipital condyle than the right crest (Fig. 7), the cranium underwent some degree of differential dorsoventral crushing. Such a slight crushing may have resulted in minor changes of orientation for processes and crests.

Ontogenetic stage

Considering the closure of all cranial sutures, including the complete ankylosis of the dorsomedial suture between premaxillae on the rostrum, the complete ankylosis of the epiphyses of the post-axis cervical vertebra and limb bones (see Galatius & Kinze, 2003), the robust aspect of cranial bones and crests, and the extensive occlusion wear facets in all preserved upper teeth (see below), the holotype CZA 1-16 is interpreted as sub-adult to adult.

General morphology

Estimated postorbital and bizygomatic widths indicate a cranial width between 280 and 290 mm (Table 1); this is in the upper part of the range for Inia geoffrensis (see measurements in Pilleri & Gihr, 1979; O Lambert, pers. obs.), whose body length reaches up to 2.50 m in adult males (Martin & Da Silva, 2006). The rostrum is wider than high at its preserved anteriormost portion; it widens markedly towards the lost antorbital notch. The anterior part of the facial region is much wider than the posterior part, due to the abrupt narrowing of the maxilla and frontal towards the nuchal crest, leaving the temporal fossa nearly completely dorsally open (Fig. 4), as in I. geoffrensis. The temporal fossa is anteroposteriorly long, extending posteriorly far beyond the medial region of the supraoccipital shield due to the posterior projection of the temporal crest until the level of the occipital condyles (Fig. 5). The elevation of the facial region towards the anteroposteriorly long vertex is moderate, but abrupt, with a vertex considerably higher than in the extant pontoporiid Pontoporia blainvillei and some fossil relatives (e.g., Pliopontos littoralis and Pontistes rectifrons Bravard, 1858, see Muizon, 1984; Lambert & Muizon, 2013). The nuchal crest is high and greatly thickened. The nuchal crest tends to be proportionally thicker in older individuals (identified by the greater degree of closure of the cranial sutures) of I. geoffrensis from our sample.

Table 1 Measurements (in mm) of the cranium of the iniid Kwanzacetus khoisani CZA 1-2 (holotype), late Miocene of Angola.

Width of rostrum at base	+130	
Height of rostrum at base	+65	
Width of premaxillae at anterior margin of right premaxillary foramen	67	
Width of bony nares	e49	
Width of premaxillary sac fossae	e89	
Width of right premaxillary sac fossa	e35	
Width of left premaxillary sac fossa	36	
Postorbital width	e281	
Bizygomatic width	e286	
Maximum anteroposterior length of right temporal fossa	155	
Height of right temporal fossa from floor of squamosal fossa to top of temporal crest	103	
Minimum posterior distance between maxillae across vertex	25	
Distance between temporal fossae at level of medial part of nuchal crest	121	
Minimum posterior distance between temporal crests	e91.5	
Width of occipital condyles	e96	
Height of right occipital condyle	54	
Width of foramen magnum	e44	
Notes.

e estimate

+ incomplete

Premaxilla

At the anterior end of the rostrum as preserved, which is at about 300 mm from the anterior boundary of the bony nares, the two premaxillae are dorsomedially ankylosed, with no trace of suture left; at this level the cross section of the premaxillae is semi-circular, as in Brujadelphis ankylorostris and Isthminia panamensis (Pyenson et al., 2015; Lambert et al., 2017). A clear separation of the premaxillae only occurs a short distance (32 mm) anterior to the bony nares (Fig. 4). Preserved margins of the damaged bone in this area suggest that the anterior outline of the nares was originally more U-shaped than V-shaped. At rostrum base, the dorsal surface of each premaxilla is slightly transversely convex, facing dorsolaterally. Visible on both sides, the premaxillary foramen is roughly at the same anteroposterior level as the position of the antorbital notch (estimated based on the outline of the lacrimojugal complex, as compared to other delphinidans), or only slightly posterior. Each foramen is followed anteriorly by a shallow anteromedial sulcus for at least 35 mm. Lateral to the sulcus, the premaxilla is considerably thicker, more convex, whereas in the prenarial triangle the dorsal surface is slightly transversely convex and subhorizontal, with the medial margin barely more elevated. The posterolateral sulcus is shallow for most of its extent, reaching at least the level of the anterior margin of the bony nares (surface damaged more posteriorly), and the posteromedial sulcus is indistinct. The surface of the premaxillary sac fossa is moderately convex, both transversely and anteroposteriorly, and thickened, with a maximum height above the lateral premaxilla-maxilla suture of 9.5 mm on the left side; such a condition corresponds to a low premaxillary eminence, lower than in adults of Inia geoffrensis and, to a lesser extent, than in the holotype of Brujadelphis ankylorostris. The lateral margin of the premaxilla is straight in the rostrum base region, with a moderate divergence of right and left margins until the level of the anterior margin of the bony nares. The posterior end of each premaxilla is damaged and incomplete. On both sides a thin layer of sediment covers part of the maxilla along the posterolateral margin of the bony nares, without any trace of premaxilla (Figs. 4, 9); this indicates that the premaxillae originally most likely did not reach the posterior margin of the nares, only made by the maxillae, and diverged somewhat posterolaterally, as in some inioids (e.g., Inia geoffrensis and Pontoporia blainvillei). A more complete specimen will be necessary to more precisely assess the shape and extent of the posterior end of each premaxilla. In lateral view the posterodorsal elevation of the premaxilla starts at the level of the premaxillary foramen, but it remains weak until the preserved posterior end of the premaxilla, the lateral margin of the bony nares being much lower than the top of the nasal. The dorsolateral premaxilla-maxilla suture is closed on the rostrum, and it lacks any lateral groove, a condition differing from Pontoporia blainvillei.

Maxilla

At the preserved anterior end of the rostrum the maxilla is exposed lateral to the premaxilla in dorsal view (Fig. 4). At rostrum base, the dorsal exposure of the maxilla has roughly the same width as the premaxilla, forming a wide and slightly transversely concave surface. On the left side, only two small dorsal infraorbital foramina opening anterolaterally could be detected, just behind the level of the premaxillary foramen; the anteromedial foramen has a diameter of 5 mm and the posterolateral a diameter of more than 4 mm. No other foramen is visible on the dorsal surface, but surfaces are damaged and partly covered with sediment. Posterior to the postorbital process the lateral margin of the maxilla is directed posteromedially, more than in Brujadelphis ankylorostris and Isthminia panamensis, and conspicuously elevated dorsolaterally. This elevated margin defines posterolaterally a deep fossa, limited posteriorly by the high nuchal crest and medially by the frontal and nasal on the vertex (Figs. 4, 9). This deep fossa is reminiscent of B. ankylorostris, Ischyrorhynchus vanbenedeni, Inia geoffrensis, and Isth. panamensis, but also Hadrodelphis calvertense Kellogg, 1966 and Liolithax pappus (Kellogg, 1955). The medial part of the maxilla is vertical along the vertex; on the right side the medial margin even slightly overhangs the underlying part of the bone (Fig. 9). Included in the nuchal crest the posterior margin of the left maxilla is regularly rounded, lacking any posterolateral and posteromedial angles, whereas a more conspicuous posteromedial angle is present on the right side.

On the preserved anterior part of the rostrum the ventral surface of the maxilla is roughly flat, becoming gradually more transversely convex posteriorly, until the level of the anterior margin of the lacrimojugal complex, where it is markedly convex (Figs. 6, 8). The preserved anteriormost alveoli are located just medial to the lateral margin of the maxilla; they have a transverse diameter of 10–11 mm and are separated by septa about 10 mm long. The right ventral infraorbital foramen is preserved, but the contributions of surrounding bones are not clear; it is probably margined by the frontal posterolaterally and by the lacrimojugal complex anterolaterally.

Palatine and pterygoid

Only the anteromedial part of the maxilla-palatine suture is visible, extending anterolaterally until a level 68 mm beyond the anterior limit of the pterygoid sinus fossa (Figs. 6, 8). Apices of right and left palatines are thus separated by a stripe of maxillae and/or vomer, at least 17 mm wide anteriorly. The ventrolateral surface of the palatine in the rostrum base is slightly transversely concave and crossed by a series of thin longitudinal grooves and crests, the latter most likely corresponding to the suture with the lost pterygoid. In this region, the pterygoid seems to be preserved only as a small scale of bone on the anteroventral corner of the anteriorly short pterygoid sinus fossa. Located posterior to the approximate level of the antorbital notch and about 25 mm anterior to the choanae, the anterior margin of this fossa forms a straight, laterally directed crest that turns posterolaterally 18 mm medial to the ventral infraorbital foramen. From this anterolateral corner, the pterygoid sinus fossa deepens distinctly posterodorsally, in the region just anterolateral to the choana. There, the lateral wall of the fossa is made of the partly preserved, thin lateral lamina of the palatine. Small pieces of the medial lamina of the pterygoid are preserved lateral to the choana and towards the basioccipital basin.

Vomer

The vomer is only visible ventrally: between the pterygoid sinus fossae, between the choanae, and as a fragmented plate in the anterior part of the basioccipital basin (Figs. 6, 8).

Presphenoid and cribriform plate

Due to the narrow gap between the premaxillary sac fossae, partly filled with sediment, the presphenoid is only visible at the nasal septum (Figs. 4, 9). The latter is acute and reaches a high level on the cribriform plate. Whereas the dorsal edge of the plate is visible in dorsal view, some degree of anterodorsal projection of the nasals gives the plate an anterodorsal direction, slightly overhanging the bony nares and thus hiding the anterior surface of the plate in dorsal view.

Nasal

Although the sutures of the nasals with surrounding bones are difficult to follow, the general shape of these bones can be described. Each nasal is markedly wider anteriorly; as wide as the bony nares in their anterior region, both nasals abruptly narrow posteriorly towards the posterior contact with the frontal (Figs. 4, 5, 9). Related to this strong posterior narrowing, each nasal has a much longer surface of contact with the corresponding maxilla than with the corresponding frontal. The convex dorsal surface of each nasal slopes laterally and slightly anteriorly. The anteromedial part of this surface is somewhat abraded, but a shallow and narrow internasal fossa may have been originally present. The anterior margin of each nasal is only slightly anteriorly concave. The nasals of CZA 1-16 are considerably reduced, anteroposteriorly shorter compared to Atocetus spp., Brujadelphis ankylorostris, Isthminia panamensis, and Liolithax pappus, but not as short as in Inia geoffrensis (Fig. 10), Ischyrorhynchus vanbenedeni, and to a lesser extent Meherrinia isoni.

Figure 10 Dorsal view of the cranium of the extant iniid Inia geoffrensis (Amazon river dolphin).

Photograph of the facial region of the cranium of I. geoffrensis (MUSM DPV CE 9) in dorsal view, showing several morphological features shared with Kwanzacetus khoisani and, for part of them, with other inioids and some early delphinidans (see text for details). Photo credit: Giovanni Bianucci.

Frontal

Narrow and relatively low at their contact with the nasals on the vertex, the frontals widen and thicken posteriorly, reaching a maximum height at about mid-length of their dorsal exposure and a maximum transverse width (29 mm) close to their posterior margin (Figs. 4, 5, 7, 9). Markedly higher than the nasals and nuchal crest, this prominent highest part of the frontals (frontal boss sensu Muizon, 1988) has a semi-circular outline in dorsal view, much similar to the condition in Inia geoffrensis (Fig. 10) and Ischyrorhynchus vanbenedeni; the dorsal surface is anteroposteriorly convex, with lateral margins overhanging the underlying maxilla (best seen in anterodorsal and posterodorsal views). The posterior margin of the prominence is defined by an abrupt step towards the nuchal crest; these two regions are separated by a shallow transverse groove, in a way similar to I. geoffrensis (with the groove being anteriorly convex in the latter). It should be noted that the frontal boss tends to be transversely wider and more prominent in older individuals of I. geoffrensis from our sample.

None of the preorbital processes of the frontal is preserved, and the lateral margin of the orbit is lost on the right side. The long (35 mm) and robust right postorbital process is directed ventrally, slightly posteriorly, and distinctly laterally, with its lateral surface widely exposed in dorsal view (Figs. 4, 9), as in I. geoffrensis. The cross section of this process is triangular, related to the development of a high infratemporal crest along its ventromedial surface, reaching its apex (Figs. 6, 8). Just posterior to the infratemporal crest, the ventral surface of the frontal is excavated by a large and deep fossa in its medial part. Extending laterally without any defined lateral and posterolateral boundaries, this fossa is interpreted as for a large postorbital lobe of the pterygoid sinus. This lobe is comparatively large in I. geoffrensis, and smaller in Pontoporia blainvillei (Fraser & Purves, 1960). Anteromedial to the narrow, triangular medial portion of the frontal groove, another large and deep fossa is observed, presumably for an extended preorbital lobe of the pterygoid sinus. This fossa is posterolaterally margined by a high, thin plate of bone (see orbitosphenoid below). More laterally, depressions in the ventral surface of the frontal are less clearly defined.

Lacrimojugal complex

The only preserved part of the right lacrimojugal complex is anterolateral and lateral to the ventral infraorbital foramen (Figs. 6, 8). There, an oblique lacrimal-frontal suture can be observed. A low oblique crest directed anteromedially along the preserved lateral margin of the antorbital region may correspond to the base of the styliform part of the jugal.

Orbitosphenoid

The medialmost part of the thin plate of bone separating the frontal groove from the fossa for the preorbital lobe of the pterygoid sinus is most likely made of the orbitosphenoid (Fig. 8).

Alisphenoid

With an anteroposterior diameter of 9 mm, the foramen ovale is large (Fig. 8). It is separated from the posterior lacerate foramen by a thick bony bridge. Its anterior wall is dorsoventrally thick and anteroposteriorly narrow, followed anteriorly by a wide, only slightly concave surface, facing ventrally and slightly anterolaterally. The damaged lateral part of this surface was probably somewhat more concave, being laterally margined by an elevated subtemporal crest. The latter only extends anteriorly until a level at about mid-length of the orbital fissure.

Parietal and interparietal

The nuchal crest being anteroposteriorly thick for its whole extent (minimum thickness on right side = 25 mm; on left side = 22 mm) (Figs. 4, 5, 9), a part of it could be made of the parietals (although the frontal is markedly thickened along the nuchal crest in Inia geoffrensis; Fig. 10). However, all sutures are obscured in this region. Nevertheless, a small but well-defined prominence located along the sagittal axis, between the frontal boss and the posterior wall of the nuchal crest, probably corresponds to part of an interparietal. A similar prominence was observed in part of the I. geoffrensis crania analyzed (e.g., USNM 239667, 395415, 49582; ZMA 17771; MUSM DPV CE 9; Fig. 10).

Supraoccipital

The posterior wall of the nuchal crest overhangs the more ventral part of the supraoccipital, and the crest reaches a height of 21 mm in its medial part. Whereas on the right side this posterior margin is posteriorly concave and draws a regular curve, on the left side the anterolateral corner displays a distinctly posteriorly convex margin (Figs. 7, 9). A similar asymmetric occipital tuberosity was observed on the same side in several crania of Inia geoffrensis (e.g., MUSM DPV CE 9; USNM 239667; Miller, 1918, pl. 3; Van Beneden & Gervais, 1880, pl. 33; Fig. 10; and to a lesser extent USNM 395415); the absence of this feature in other crania of I. geoffrensis was not found to be related to a different ontogenetic stage, based on the sample available for this study. As mentioned above (general morphology) the high temporal crests project far posteriorly, 30 mm more than the median region of the supraoccipital shield; the crests display roughly the same extent in Ischyrorhynchus vanbenedeni MLP 5-16 (Pilleri & Gihr, 1979, pl. 6), are shorter in Brujadelphis ankylorostris, I. geoffrensis, and Isthminia panamensis, but longer in the large early delphinidans Hadrodelphis calvertense and Liolithax pappus. In posterior view right and left crests are roughly parallel for most of the height of the shield, and are relatively close to each other, reaching medially the lateralmost margin of the premaxilla and making thus the supraoccipital shield much narrower than the supraorbital region of the cranium. Between the crests the supraoccipital is dorsoventrally convex, only excavated by a shallow sagittal groove starting 40 mm above the foramen magnum and extending dorsally for more than 32 mm. Just below the nuchal crest a short, thin, but relatively high (6 mm) external occipital crest is observed, as in I. geoffrensis. Dorsolateral to each condyle, the dorsal condyloid fossa is shallow.

Exoccipital

The occipital condyles are separated from the neurocranium by a moderately developed condylar neck (16 mm long on the right side). Lateral to the condyle the posterior surface of the exoccipital is overhung by the prominent temporal crest (Fig. 7). Directed ventrolaterally and slightly posteriorly this surface does not reach the posterior level of the occipital condyles. Ventrally, the articulation surface for the stylohyal on the paroccipital process is a concave surface, anteroposteriorly longer than wide. Anterolateral to this feature, the anterior surface of the exoccipital is marked by a wide, ventrolaterally directed groove most likely corresponding to the paroccipital concavity (for the posterior sinus) (Fig. 8). Leading dorsomedially to a deep, bowl-shaped depression that is well separated from the posterior lacerate foramen, the groove extends also ventroposteriorly on the ventral margin of the exoccipital. The jugular notch is deeper than wide, with a minimum ventral width of 10.5 mm and a depth of 15 mm.

Basioccipital

Right and left basioccipital crests diverge markedly posterolaterally, defining a broad basioccipital basin (Figs. 6, 8); this condition may have been somewhat accentuated due to some degree of dorsoventral crushing, but it is similar to what is observed in Inia geoffrensis (as compared for example to the narrower basin in Delphinus delphis Linnaeus, 1758, not reaching laterally beyond the lateral margins of the occipital condyles). The crests are transversely thin in their anterior portion. Only preserved on the right side, a marked thickening occurs a short distance from the jugular notch; there the ventromedial surface of the crest bears a protuberance that is posteriorly defined by a transverse crest; the latter runs for a short distance (20 mm) dorsomedially, followed towards the floor of the basioccipital basin by a slight bulge of the surface. The lateral surface of the crest is excavated in a dorsomedial direction by a wide groove, leading to a large elliptical fossa with a maximum diameter of 18 mm, medial to the posterior lacerate foramen. The position of this fossa relative to the posterior lacerate foramen and the foramen ovale suggests that it most likely contains the ventral carotid foramen, although it probably corresponds to a diverticulum of the peribullary sinus surrounding this foramen, as seen for example in D. delphis.

Squamosal

Whereas the left squamosal is not preserved, the right zygomatic process is nearly completely lost; considering the anteroposterior level of the postorbital process of the frontal and the long temporal fossa, this process was most likely elongated. The mandibular fossa is wide (more than 35 mm), nearly transversely flat, and facing anteroventrally (Figs. 6, 8). The medialmost part of the postglenoid process is a thin, transversely directed crest. The preserved section of the lateral part of the process indicates that it was considerably anteroposteriorly thicker. Medial to the mandibular fossa the large tympanosquamosal recess is triangular in outline. The recess may have extended posterolaterally, posterior to the postglenoid process, as a deep and narrow groove, as for example in Brujadelphis ankylorostris and Liolithax pappus CMM-V-3780. However, this area is poorly preserved and the external auditory meatus cannot be outlined. The falciform process of the squamosal is reduced to a low, oblique crest medial to the tympanosquamosal recess; this crest is only slightly swollen at the anteroposterior level of the foramen ovale. From this level, the squamosal extends for a short distance anteriorly as a thin plate. This plate being incomplete, a contact with a lost lateral lamina of the pterygoid (as in Pontoporia blainvillei) cannot be completely excluded.

The posttympanic process of the squamosal is characterized by a deeply concave, posterolaterally facing surface, overhung by a thick prominence of the supramastoid crest. A similar prominence is observed in various early diverging delphinidans, including B. ankylorostris, Inia geoffrensis, L. pappus, and Macrokentriodon morani Dawson, 1996 (see Lambert et al., 2017).

The squamosal fossa is extremely wide (Figs. 4, 9); the distance from the medial surface of the temporal fossa to the lateralmost margin of the squamosal fossa is at least 64 mm. The floor of the fossa is transversely concave and slightly anteroposteriorly concave for the anterior two thirds of its length. No deep depression is observed in the fossa, different from the condition in L. pappus CMM-V-3780.

Upper teeth

Five upper teeth are partly preserved, including two in situ in the right maxilla at about 160–190 mm anterior to the level of the antorbital notch and three in a detached, more posterior fragment of the right maxilla (Fig. 11). The maximum transverse diameter of the robust crown ranges from 8.7 to 9.6 mm in the anterior teeth to 9.5 to 10.5 mm in the posterior teeth. In one of the better preserved anterior teeth the height of the crown only reaches 11.4 mm, meaning that these crowns are not significantly longer than wide. A long (26 mm), posterodorsally directed root is preserved for one of the posterior teeth. The enamel on the crown of all the teeth is covered with deep longitudinal crests and grooves (wrinkled enamel); this ornamentation is more similar to Inia geoffrensis and Ischyrorhynchus vanbenedeni (see MLP 5-18), stronger than in the few other early diverging delphinidans displaying some ornamentation (e.g., Brujadelphis ankylorostris and Isthminia panamensis). The lingual side of the crown of all the preserved teeth lacks any heel, a clear difference with I. geoffrensis, and no accessory denticles are observed. Although the preservation state is not optimal, the five teeth display a crown that is truncated along its mesial to mesolingual side: from a relatively shallow occlusion facet in one posterior tooth to the removal of up to half the crown in several other teeth, a condition that is also observed for example in Liolithax pappus USNM 15985, but not in the studied specimens of I. geoffrensis. This pattern indicates extensive attritional (tooth to tooth) wear, at least in the posterior part of the jaws, for this individual.

Figure 11 Maxillary teeth of Kwanzacetus khoisani.

(A) two more anterior right maxillary teeth of the holotype of K. khoisani CZA 2 in lingual view; (B) detail of one of these teeth in labiodistal view; (C) three more posterior right maxillary teeth CZA 3 in lingual view; (D) detail of one tooth in labiomesial view; (E) detail of the same tooth in labiodistal and slightly occlusal view; (F) detail of another tooth in labiomesial view. Dotted lines for deep occlusion facets. Scale bars equal 10 mm. Photo credit: Olivier Lambert.

Cervical vertebrae

Axis (Figs. 12A–12D; Table 2)

Both transverse processes are incomplete and the top of the neural arch is missing. The axis was not fused to the atlas and C3. The odontoid process is short, only slightly longer anteriorly than the anterior articular facets. A sagittal keel marks the ventral surface of the centrum. The posterior articular surface is deeply concave. The base of the transverse process is dorsoventrally high and the process was originally longer than in Inia geoffrensis (see Van Beneden & Gervais, 1880; Miller Jr, 1918). The pedicle is anteroposteriorly long (minimum length 22 mm).

Figure 12 Cervical vertebrae of Kwanzacetuskhoisani.

(A–D), axis of the holotype of K. khoisani CZA 4 in anterior (A), left lateral (B), posterior (C), and ventral (D) views; E–F, cervical ?C3-C4 CZA 5 in anterior (E) and right lateral (F) views. Scale bar equals 50 mm. Photo credit: Olivier Lambert.

Table 2 Measurements (in mm) of the cervical vertebrae of the iniid Kwanzacetus khoisani CZA 4-5 (holotype), late Miocene of Angola.

Axis		
Maximum width as preserved	127	
Maximum width across anterior articular facets	e106	
Height of left anterior articular facet	e47	
Width of left anterior articular facet	e37	
Height of posterior epiphysis	e37	
Width of posterior epiphysis	e63	
Width of neural canal	33	
Maximum anteroposterior length along sagittal plane	39	
Other cervical vertebra (C3 or C4)		
Maximum width	e106	
Height of anterior epiphysis	42	
Width of anterior epiphysis	e49	
Height of posterior epiphysis	43	
Width of neural canal	37	
Height of neural canal	30	
Anteroposterior length of centrum	13.5	
Notes.

e estimate

Other cervical vertebra (Figs. 12E–12F, Table 2)

This vertebra is nearly complete, only missing the distal part of the left transverse process and the lateral part of the left pre- and postzygaphophysis. The unfused centrum is anteroposteriorly short, with a pentagon-shaped outline in anterior view. Whereas the anterior articular surface is roughly flat, the posterior surface is slightly concave. A keel is present on the ventral surface of the centrum. With a maximum length at mid-height of the centrum, the transverse process is made of a thin blade, markedly curved anterodorsally and anteroventrally (anterior surface being dorsoventrally concave). The base of the process is pierced by a medium-size vertebrarterial canal (maximum transverse diameter of right foramen 8.5 mm). The neural canal is roughly triangular, lower than the height of the centrum. The short pedicles are transversely wide and anteroposteriorly flattened. The better-preserved right prezygapophysis is a roughly flat surface, facing anterodorsally and slightly medially. The neural arch is slender, with only a low protuberance for the neural spine. Proportions and position of the transverse process and vertebrarterial canal are closer to C3 in Inia geoffrensis (see Miller Jr, 1918), but with a smaller vertebrarterial canal medial to a more extended lateral part of the transverse process. Good similarities are also noted with C4 of Pontoporia blainvillei (Van Beneden & Gervais, 1880; O Lambert, pers. obs.). This cervical is thus interpreted as a C3 or C4, pending the discovery of a more complete vertebral column of Kwanzacetus khoisani.

Phylogenetic Analysis

Similarly to the cladistic analyses by Post, Louwye & Lambert (2017), preliminary tests yielded highly volatile relationships for several Neogene inioids and other early delphinidans characterized by fragmentary type material; pending the discovery of more complete specimens, including the basicranium and ear bones, these taxa (Auroracetus bakerae Gibson & Geisler, 2009, Ischyrorhynchus vanbenedeni, Lophocetus repenningi Barnes, 1978, Meherrinia isoni, Pithanodelphis cornutus du Bus, 1872, Protophocaena minima, and Stenasodelphis russellae) were removed from the analysis leaving a set of 101 operational taxonomic units. Our final heuristic search resulted in a single most parsimonious tree (score 2025.78 steps, consistency index 0.16, and retention index 0.56; Fig. 13; Supplemental Information).

Figure 13 Phylogenetic relationships of Kwanzacetus khoisani.

Phylogenetic tree showing the relationships of K. khoisani with other early diverging delphinidans, as obtained from our parsimony analysis of morphological data, constrained with a molecular tree as backbone. Other odontocete clades are collapsed to facilitate reading. K. khoisani falls as an iniid, displaying close relationships with the extant Inia geoffrensis. Stars identify species with a strictly freshwater distribution. Numbers indicate bootstrap values.

A first point to be commented is the low support for many of the nodes of the obtained tree; indeed, except for a few nodes among inioids (Lipotidae and the genus Parapontoporia) bootstrap values are lower than 50, as in previous analyses of this clade (e.g., Pyenson et al., 2015). Such a low support should be taken as a warning for dealing with the proposed topology with caution. More complete specimens, especially with ear bones associated, are clearly needed for many taxa to obtain more reliable phylogenetic relationships. Furthermore, several taxa, including the important Ischyrorhynchus vanbenedeni, were omitted from the analysis. Keeping these important points in mind and focusing on early diverging delphinidans (complete tree in Supplemental Information), the obtained topology differs from Post, Louwye & Lambert (2017) in a series of early delphinidans (including for example Kentriodon pernix Kellogg, 1927, Delphinodon dividum True 1912, Hadrodelphis calvertense, Liolithax pappus, and Macrokentriodon morani) making a large clade branching after Lipotidae + Inioidea, instead of constituting successive branches of stem delphinidans (the latter result having been found in another recent study by Peredo, Uhen & Nelson, 2018). Such a monophyletic, later branching ‘Kentriodontidae’ clade was similarly found under suboptimal topologies of a previous analysis of this matrix (Lambert et al., 2017). This large stem delphinoid clade also includes Atocetus spp., recovered as pontoporiids by Post, Louwye & Lambert (2017). Among inioids, a monophyletic family Pontoporiidae is recovered, as in Lambert et al. (2017) and Post, Louwye & Lambert (2017), but differing from the analysis of Pyenson et al. (2015); in addition to the extant Pontoporia blainvillei, it includes Brachydelphis spp., Pliopontos littoralis, and Scaldiporia vandokkumi (Post, Louwye & Lambert, 2017). Interestingly, Brujadelphis ankylorostris and Isthminia panamensis are recovered as sister groups within the family Iniidae (Pan-Inia in Pyenson et al., 2015), as previously hypothesized (Lambert et al., 2017). The new taxon Kwanzacetus khoisani appears more closely related to Inia geoffrensis than these two species.

Discussion

Systematics and phylogeny

Kwanzacetus khoisani shares morphological features with several early delphinidans and inioids: a deep fossa in the maxilla between the vertex and the elevated lateral margin of the supraorbital process (e.g., Brujadelphis ankylorostris, Hadrodelphis calvertense, Ischyrorhynchus vanbenedeni, Inia geoffrensis, Isthminia panamensis, and Liolithax pappus); an anteroposteriorly long temporal fossa (e.g., B. ankylorostris, H. calvertense, Isch. vanbenedeni, I. geoffrensis, and L. pappus); the temporal crest projected posteriorly beyond the supraoccipital shield (e.g., H. calvertense, Isch. vanbenedeni, and L. pappus); and a broadly dorsally exposed squamosal fossa, more so than in lipotids (e.g., H. calvertense, Isch. vanbenedeni, I. geoffrensis, Kampholophos serrulus Rensberger, 1969, and L. pappus). It shares additional features with part of the inioids: low premaxillary eminences (also seen in several phocoenids, but not in Isth. panamensis) and the premaxilla most likely not contacting the nasal (also seen in some lipotids and many delphinoids, but not in Brachydelphis spp. and Pontistes rectifrons, see Post, Louwye & Lambert, 2017).

Among inioids, it shares one additional character with B. ankylorostris and Isth. panamensis: partial ankylosis of the premaxillae on rostrum. It shares with I. geoffrensis and Isch. vanbenedeni: the combination of a frontal boss with nasals being lower than the frontals on the vertex (also seen phocoenids); and the proportionally robust teeth being markedly ornamented, with wrinkled enamel. Finally it shares with I. geoffrensis: the laterally directed postorbital process of the frontal (unknown in Isch. vanbenedeni); the anteroposterior thickening of the nuchal crest (even more than in the latter); the more developed left occipital protuberance (only observed in some specimens of I. geoffrensis) (Fig. 10).

The attribution of Kwanzacetus khoisani to the family Iniidae and its close relationships with I. geoffrensis are also found in our phylogenetic analysis, although with low support. Additional morphological information on the new taxon, for example on the ear bones (not preserved in the holotype) will potentially bring further support to our hypothesis. Furthermore, additional information on Isch. vanbenedeni, through direct observation of the known specimens and the discovery of more complete material, would allow testing the hypothesis (based on the comparison of facial features) that this important iniid species from freshwater deposits of South America is more closely related to I. geoffrensis than K. khoisani.

This new inioid genus and species further increases the inioid diversity during the late Miocene, a time interval confirmed here as by far the heyday for this superfamily. Indeed, a vast majority of the extinct inioids is found in Tortonian and Messinian deposits (e.g., Cozzuol, 2010; Gutstein, Cozzuol & Pyenson, 2014; Pyenson et al., 2015, fig. 15; Marx, Lambert & Uhen, 2016; Murakami, 2016; Post, Louwye & Lambert, 2017; Di Celma et al., 2017).

Palaeoecology and palaeobiogeography

The morphological similarities of this presumably marine species with the two known freshwater iniids (the extant Inia geoffrensis and the late Miocene Ischyrorhynchus vanbenedeni) at the level of the dentition (robustness and ornamentation of teeth), size and outline of the temporal fossa, and extent of the temporal and nuchal crests suggests at least some degree of functional anatomy overlap, for prey types, feeding strategies, and presumably locomotion. Note however the deep occlusion facets and lack of any heel on the crowns of posterior maxillary teeth in Kwanzacetus khoisani, two significant differences with I. geoffrensis. In this context of potential partial ecological overlap, the asymmetric development of the left occipital protuberance observed in both I. geoffrensis and K. khoisani may indicate a similar degree of behavioral motor asymmetry (or laterality), a phenomenon reported in many cetaceans (review in Platto et al., 2017). Interestingly, although no clear correlation has been demonstrated with laterality, four species of freshwater dolphins, including I. geoffrensis, have been reported performing side-swimming (Renjun et al., 1994; Platto et al., 2017). The preference for one side during swimming and/or feeding may explain the presence of a larger left occipital protuberance, for stronger neck muscles (M. semispinalis capitis or rectus capitis posterior major), presumably also present in the last common ancestor of I. geoffrensis and K. khoisani.

The record of a close relative of I. geoffrensis in marine deposits from the eastern coast of South Atlantic further corroborates the hypothesis that part of the early evolution of the lineage of the latter (after the split with Pontoporiidae) occurred in the marine environment (Pyenson et al., 2015). Alternatively, Kwanzacetus khoisani may represent a less likely ecological reversal to a marine habitat, following the middle to early late Miocene colonization of freshwater habitats by iniids in South America (as proposed for another late Miocene inioid from Panama; Pyenson et al., 2015). The oldest iniid remains from freshwater deposits of South America indeed come from late Miocene levels of Entre Rios province (Paraná Basin, Argentina) and Acre state (Brazil) localities (Cozzuol, 2010; Gutstein, Cozzuol & Pyenson, 2014). A better-constrained chronostratigraphic context for the holotype of K. khoisani and for South American iniid fossils would provide a more precise temporal framework.

Interestingly, three other late Miocene marine relatives of I. geoffrensis (Brujadelphis ankylorostris, Isthminia panamensis, and Meherrinia isoni) were discovered in different geographic areas (southeast Pacific, Carribean Sea, and North Atlantic; Geisler, Godfrey & Lambert, 2012; Pyenson et al., 2015; Lambert et al., 2017; Post, Louwye & Lambert, 2017). K. khoisani being interpreted here as morphologically closer to I. geoffrensis than any of the three species mentioned above, this may indicate that the transition from a marine environment to the strictly freshwater Amazonian habitat of the latter did occur on the Atlantic side of South America. This hypothesis is further supported by the fact that some of the oldest freshwater iniid remains from South America were found in an area (Paraná Basin) relatively close to the Atlantic coast (Gutstein, Cozzuol & Pyenson, 2014) and that the Acre area, where other late Miocene iniid fossils were discovered, may have been connected with the South Atlantic by a large river system since about 10 Ma (onset of Amazon fan; Hoorn et al., 2010). It is also worth mentioning that the only surviving member of iniids’ sister-group, Pontoporia blainvillei (franciscana), lives along the eastern coast of South America (Brownell Jr, 1989). We anticipate that new finds from both coasts of the South Atlantic will most likely shed further light on this still poorly understood transition. In a broader context, the first description of a Neogene cetacean from inland deposits of western sub-Saharan Africa (see review in Gingerich, 2010) reveals the potential of this large coastal area for deciphering key steps of the evolutionary history of cetaceans in the South Atlantic.

Conclusions

Based on a partial dolphin skeleton discovered in marine deposits from the late Miocene (Tortonian - Messinian) of Angola, southwestern Africa, we describe a new genus and species, Kwanzacetus khoisani. The new taxon is referred to the family Iniidae, and among iniids it shares several cranial (e.g., frontal boss) and dental features (e.g., wrinkled enamel) with the extant Amazon river dolphin Inia geoffrensis. Confirmed by our phylogenetic analysis, the close relationship with the latter species suggests that iniids’ marine to freshwater transition may have occurred during the middle to late Miocene along the Atlantic coast of South America. Finally, this first neocete taxon described from the Neogene of Angola reveals the potential of the southeastern Atlantic area for elucidating some crucial stages of cetacean evolutionary history.

Supplemental Information

Supplemental Information 1 Complete phylogenetic tree resulting from our analysis

Click here for additional data file.

Supplemental Information 2 Character-taxon matrix used for the phylogenetic analysis

Click here for additional data file.

Supplemental Information 3 Molecular tree used as a backbone constraint

Click here for additional data file.

We wish to thank Sébastien Bruaux (IRSNB, Brussels, Belgium), Stephen J. Godfrey and John R. Nance (CMM, Solomons, USA), Christian de Muizon and Christine Lefèvre (MNHN, Paris, France), Rodolfo Salas-Gismondi, Mario Urbina and Rafael Varas-Malca (MUSM, Lima, Peru), Henry van der Es (NMR, Rotterdam, The Netherlands), David J. Bohaska, Charles W. Potter, and Nicholas D. Pyenson (USNM, Washington DC, USA) for access to collections under their care; Giovanni Bianucci (Università di Pisa, Italy) for providing photos of skull material of Ischyrorhynchus vanbenedeni at MLP for comparison and a dorsal view of a skull of Inia geoffrensis at MUSM; Giovanni Bianucci, Christian de Muizon, and Klaas Post for fruitful discussions on various aspects of inioid evolution; Etienne Steurbaut (IRSNB, Brussels, Belgium) and Stephen Louwye (Universiteit Gent, Belgium) for testing the microfossil content of sediment samples associated to the dolphin skull; Eddy Metais, Christian Seyve, and Tatiana Tavarez (Total Exploration Production Angola, Luanda, Angola), António Olímpio and Neuza Mulanda (Agostinho Neto University, Luanda, Angola) for their help during fieldtrips and logistics; Sébastien Enault (ISEM Montpellier - Kraniata.com, France) for constructive comments on a first draft of the paper; the reviewers Mario Cozzuol, Carolina Gutstein, and Nicholas D. Pyenson, and the editor Tomas Hrbek for the detailed reviews that significantly improved the quality of this work.

Institutional Abbreviations

CMM Calvert Marine Museum, Solomons, Maryland, USA

CZA Universidade Agostinho Neto, Luanda, Angola (CZA in reference to the Cuanza River)

IRSNB Institut Royal des Sciences Naturelles de Belgique, Brussels, Belgium

ISEM Institut des Sciences de l’Evolution Montpellier, France

MGUH Geological Museum, University of Copenhagen, Copenhagen, Denmark

MLP Museo de La Plata, La Plata, Argentina

MNHN Muséum National d’Histoire Naturelle, Paris, France

MUSM Museo de Historia Natural, Universidad Nacional Mayor de San Marco, Lima, Peru

NMB Natuurhistorisch Museum Boekenberg, Antwerp, Belgium

NMR Natuurhistorisch Museum Rotterdam, Rotterdam, The Netherlands

TM Teylers Museum, Haarlem, The Netherlands

USNM National Museum of Natural History, Smithsonian Institution, Washington, D.C., USA

ZMA Zoölogisch Museum Amsterdam, The Netherlands

Additional Information and Declarations

Competing Interests

Author Contributions

Data Availability

New Species Registration

The authors declare there are no competing interests.

Olivier Lambert conceived and designed the experiments, performed the experiments, analyzed the data, contributed reagents/materials/analysis tools, prepared figures and/or tables, authored or reviewed drafts of the paper, approved the final draft.

Camille Auclair conceived and designed the experiments, performed the experiments, analyzed the data, contributed reagents/materials/analysis tools, approved the final draft.

Cirilo Cauxeiro performed the experiments, analyzed the data, contributed reagents/materials/analysis tools, approved the final draft.

Michel Lopez performed the experiments, analyzed the data, contributed reagents/materials/analysis tools, prepared figures and/or tables, authored or reviewed drafts of the paper, approved the final draft.

Sylvain Adnet conceived and designed the experiments, performed the experiments, analyzed the data, contributed reagents/materials/analysis tools, prepared figures and/or tables, approved the final draft.

The following information was supplied regarding data availability:

The character-taxon matrix for phylogenetic analysis is provided in the Supplemental Files.

The following information was supplied regarding the registration of a newly described species:

Publication LSID: urn:lsid:zoobank.org:pub:9488E279-A53A-4E7A-A2F7-AF8C693C208A;

Kwanzacetus: urn:lsid:zoobank.org:act:A9919C85-25B8-4D43-8C9B-9C2DC0185599;

Kwanzacetus khoisani: urn:lsid: zoobank.org:act:09A29C2F-1CF1-45CA-9944-0AFE97759D21.

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
