# Peer review of "A close relative of the Amazon river dolphin in marine deposits: a new Iniidae from the late Miocene of Angola"

_PeerJ, doi:10.7717/peerj.5556_

## Round 0.1 · original submission · Major Revisions

Dear Authors,

I have received three reviews now. First I would like to congratulate the authors on what appears to be an important contribution to the systematics and taxonomy of Inioidea, and the implications of this study for the colonization of freshwater habitats. But there are still a number of issues that need to be addressed.

My main concern and the one of the main reason for inviting Dr. Pyenson to serve as a reviewer of this MS was that I did not see why Isthminia panamensis was not included in this analysis. The data for Isthminia panamensis are available, apparently are compatible with your data, and based on what you say in the Acknowledgments you were at the Smithsonian and so presumably you examined Isthminia panamensis. So why was this species not included? The conclusion that Kwanzacetus adamsi is sister to Inia geoffrensis is without basis if Isthminia panamensis is not included in your analyses (Isthminia panamensis is sister to Inia geoffrensis in Pyenson et al. 2015).

Otherwise I would like you to improve the photos/images so that characters and characters states are more clearly seen and phylogenetic analyses. With respect to the coding of characters as brought up by Dr. Cozzuol, I agree that character states and coding of characters need to be more explicitly justified. Also given that you are using transformation matrixes in your analyses, these need to be provided, including any constraints and constraint topologies that were used to generate the final results.

Once you address the above points and those of the referees, all of whom spent considerable amount of time reviewing this MS, I believe this MS will be a solid contribution to our understanding of the evolution Inioid dolphins and how and when South American freshwater habitats were colonized.

I look forward to your revision.
Sincerely,

Tomas Hrbek

·

Basic reporting

Lambert et al. provide a valuable contribution with a description of a new genus and species of imiid from Miocene marine rocks of coastal Angola. The fossil record of Iniidae (a group solely represented by the extant Amazon River dolphin, Inia geoffrensis) consists of a variety of fossil forms recovered from both within and outside of the Amazon Basin. In some cases, extinct relatives more closely related to Inia than any other odontocete (living or fossil) have been recovered well outside Amazonia, including North and Central America. In this way, the discovery of an Inia relative from Africa is important: its phylogenetic relationship, geographic, and geological provenance all bear on the origins and evolution of Inia’s broader group.

The manuscript is well written. The introduction is fine, and the manuscript is characteristically crisp and strong in its descriptive sections — there are no real issues there. I think a Specimens Observed section is warranted, especially given the mention of an Inia study sample — it’s also useful for specialists to know what the authors saw, especially firsthand versus via literature or shared imagery among colleagues. It’s one thing to pick this up or infer it from the Acknowledgments; it’s better to see it early on in the manuscript. Also, I am not particularly a fan of bullet item conclusion sections — is it really that hard to write it as a paragraph? In some cases, I am not convinced the perfunctory nature of a conclusion section is warranted, but I think it is fine to have one here.

Experimental design

The manuscript is weakest both content-wise and analytically with its phylogenetic analysis. The authors do not report any sub-clade support values (Bremer nor bootstrap), so there is really no basis for knowing about how well each sub-clade is supported. I think this is a real issue with this manuscript because there is a lack of stability in so-called “family Iniidae” sub-clades. The authors completely side-step any of the issues raised in Pyenson et al. (2015, PeerJ) about Cozzuol’s Ischyrorhynchinae, nor any mention of Saurocetes spp., and only one mention of Goniodelphis. Yes, Pyenson et al. (2015) used a different matrix for their analysis of Isthminia, but I would really like to see the authors make some effort to discuss the different topologies and memberships of sub-clades within Iniidae and Inioidea from previous work. It wouldn’t take much, and the recovery of a monophyletic Pontoporiidae, along with a different composition to a stem-based Iniidae, is valuable to specialists. A paragraph of comparative systematics would greatly enhance the utility of this paper, beyond a mere description of a new fossil Inia relative.

The biggest issue with the phylogenetic analysis is the decision to exclude Isthminia panamensis from the character taxon matrix and phylogenetic analysis. Why? This taxon was published in 2015, in PeerJ, with a full description; 3D models and CT datasets were released publicly with the paper. It is completely available for coding and integration. The authors spend a fair amount of space in the comparative morphological sections with Isthminia, along with Brujadelphis — the latter being a taxon published in 2017, a seeming sister taxon to Isthminia, although it is impossible to know because Isthminia wasn’t included in the phylogenetic analysis of that paper either. Unless it is explicitly stated, I don’t see a reason for excluding Isthminia from an all-inioid phylogenetic analysis and urge the authors to include it as a part of their analysis in their revisions. It would really help move things forward, rather than leaving that integration as an outstanding problem.

Validity of the findings

The manuscript does a good job in its end sections of identifying the significance of a putative Inia sister taxon from the late Miocene of Africa, but if we don’t know the relationships among Isthminia, Brujadelphis, Ischyrorhynchus and Kwanzacetus, then there really is no point in presenting any potential evolutionary hypothesis for how Iniidae evolved. Overall, I find the manuscript generally well worth publishing in PeerJ, but hope that the authors can address outstanding concerns (identified above) in their revisions.

Additional comments

Some detailed comments:

- l. 21, “Thanks” is not really the appropriate word here; it implies that freshwater invasions in odontocete history happened _because_ they could echolocate, when in reality we don’t know why these invasions happened. There are potentially many, unrelated reasons (or common ones). In essence, the sentence joins potentially unrelated ideas. I suggest stripping echolocation from the argument.

- l. 59, It’s not exactly clear to me that a broad geographic range equals evolutionary success, specifically in the case of Inia versus Lipotes or Platanista. In other words, it’s a bit arbitrary to focus on geographic range instead of say, stratigraphic duration, or molecular divergence time. Tying success to any of these factors is an arbitrary decision.

- ll. 69-71, Again connecting two ideas that are not necessarily related: vertex morphology and ecological transitions — why are these linked? Yes, I understand that the vertex of Ischyrorhynchus is very similar to Inia’s — so what? What does that particular complex mean for the timing or mode of fresh-marine transitions? I suggest editing for clarity.

- l. 81, Materials and Methods section could use a Specimens Observed section.

- ll. 118-119, I presume X and Y are lat/lon coordinates; please harmonize for consistency with geographic reporting standards by providing cardinal directions for each value.

- l. 122, “south” should not be capitalized here.

- l. 143, “supposed” is not correct, either delete or replace with “likely”

- l. 151, “evocated” is not the correct verb here; substitute “described”

- ll. 163-164, There are no rules (and PeerJ enforces none), but it is confusing to list a ranked taxonomic hierarchy with intercalated with clades names — this is why, for instance, some authors describing new taxa choose to harmonize by listing only clade names (either stem- or node-based ones, following phylogenetic taxonomy). The authors clearly get the meaning of the clade names, but the lack of rank poses a problem that is not easily solved. For intellectual consistency, I strongly advocate either deleting these two clade names, or completely de-ranking all taxonomic names. Opting for a middle ground really makes no sense. See Joyce et al. 2004 J Paleo for more; and Pyenson et al. 2015 PeerJ for the issues with imposing higher taxonomic names on monospecific ranks, such Iniidae (directly relevant to this manuscript). This latter problem is exactly why Pyenson et al. (2015) did not assign Isthminia to family Iniidae and instead used a pan stem definition of Inia that defined the clade in a stable way that could accommodate future discoveries, such as this one (see l. 192)

- l. 182, again X and Y coordinates are not standard reporting, please control and identify the relevance of “GPS Bc65”

- l. 185, italicize the ichnogenus Ophiomorpha

- l. 188, it would perhaps be worth the authors’ time to explain a bit more about Douglas Adams’ relevance or connection to cetaceans, especially a fossil one from Angola — in general, don’t assume too much about a reader’s background unless the intent is to vague and cheeky (and, in which case, recognize that can bother some readers).

- l. 190, hopefully the Table with measurements will be inserted prior to this point in the manuscript, but the author assert a large size without quantification or comparison. I’m not sure it’s relevant in the differential diagnosis.

- l. 192, a nuance: Pyenson et al. (2015) did not assign Isthminia to a ranked family Iniidae — that is a classification interpretation of the authors, using a Linnaean hierarchy.

- l. 199, if mentioning a study sample of Inia, please list Specimens Observed (see also l. 407).

- l. 209, “heel” is an unusual descriptive word for dental anatomy; suggest lingual cusp or keel or enlargement — although the more I consider it, it is actually apt for the morphology of posterior cheek teeth in Inia.

- l. 251, typically reported as “2.50 m”

- l. 274, typo, premaxilla (right?)

- ll. 472-473, suggest a hyphen for post-tympanic; no hyphen between adverb and subsequent verb “posterolaterally facing”

- l. 537, please list the taxa removed from Post et al. (2017)’s matrix instead of forcing the reader to make the comparison.

- ll. 541-556 and also ll. 582-584, the authors provide no support values for sub-clades, and thus it is impossible to evaluate any statements about support for relationships among key taxa in their phylogenetic analysis. I urge the authors to report Bremer decay index or bootstrap values in their revision.

- ll. 570, can you clarify what’s meant by “part of the inioids”? Or spell out which ones? Recommend avoiding single sentence paragraphs and recommend stitching this sentence to the subsequent paragraph.

- ll. 587-589, the statement that a single new genus and species confirms the late Miocene as the heyday of Inioidea is unsupported. I tend to agree in broad outlines (e.g., see the stratigraphically calibrated inioid tree Fig. 15 in Pyenson et al. [2015, PeerJ]), but it’s not a statement evinced by any data in this manuscript — can the authors offer any? Perhaps a tally of taxonomic richness through time for Inioidea? Or a count of range-throughs for the Tortonian from Fig. 15 in the Isthminia paper? But really, the statement has no teeth lacking comparative data — heyday for Inioidea…so what? Compared to what other groups? Aren’t all odontocetes (and cetaceans) experiencing a richness peak at this marine stage? I suggest moving this sentence to a way more appropriate place around l. 623

- l. 597, “ecological/functional anatomy” is conjoining terms that are not really the same thing. I understand the authors are trying to draw functional interpretations from a suite of morphological characters, but painting ecological categories and “functional anatomy” as the same thing is not really constructive or intellectually vigorous. Please be explicit about diction — ecology is not really easily inferred by ecomorphology alone (if that’s what’s meant here).

- l. 644, Generally, I am not impressed with bullet item Conclusions — to be frank, I find it lazy. Each sentence could just as easily be constructed as part of a paragraph,

- l. 692, please capitalize Etienne Steurbaut’s surname.

Fig. 2 caption, italicize Ophiomorpha.

Fig. 3 caption, italicize Ophiomorpha.

Fig. 7 caption, typo on “cranium”

·

Basic reporting

In general, the article is well written and based on current and pertinent literature. It describes a very interesting new iniid dolphin from Angola, which greatly improves our understanding of the evolution of the group.

Nevertheless, I do have some minor suggestions and questions about some terms (I am no expert on English grammar myself) to be checked for the authors (see annotated pdf, please). Also, the figures and tables are well prepared in a general manner. But as it is a description of a new species and figures need to be as clear as possible, I think they could be improved in some punctual aspects:

1) Figures 5, 7, 8 and 9 have a bit much of uneven illumination. Of course, light shadows are very important for correct interpretation of the morphology but in these images, the contrast between very bright and very dark portions are misleading and prevent observation of all structures. I recommend trying to compensate it in trough photo edition but if it cannot be accomplished please considered taking new photographs with less intense illumination in different positions.

2) there are some structures mentioned in the text that are not indicated in the figures. For example, the orbitosphenoid is not indicated in figure 8 as promised in the text. Please double check for all structures.

3) Figure 3 is very helpful, but as it comprises a lot of information it would be more intuitive to understand if the numbers of parts A and B were also indicated on part C of the figure, not relying only on color to make its correspondence. In the same way the red color of the contact on "part C" would be helpfully also noted in "part B". Also please provide the captions for all structures annotated on the sedimentary column (part C) and indicate what means the letter T on part B of the figure (is it the fossil recovering position?).

Experimental design

It is a well designed descriptive and phylogenetic tested paper. My only comment on this respect is about the comparisons with extant Inia, which is very necessary but could be improved if the authors considered the variation of shapes and structures between different specimens of the species (I. geoffrensis, in this case), especially regarding different ontogenetic stages. The specimen that is figured (fig. 10) is probably a subadult, older specimens can show much more developed structures that in some cases would significate closer similarities (ex. thick nuchal crest) and in other cases fewer similarities (ex. frontal boss development and shape of the ascending process of the maxilla).

This is even reinforced for the large size of the specimens and very well closed and fused sutures in the new species which contrasts with the open sutures of the maxilla and frontal (laterally) of the Inia specimen. Also, for future studies and comparisons, it is needed to add a catalog number to the specimen of Inia mentioned and figured.

Validity of the findings

no comments

Additional comments

Please see attached pdf for further suggestions on your very nice work!

·

Basic reporting

The specimen is a relatively well-preserved skull of an odontocete that certainly has some Inioid affinities. Description and comparison may be improved and a broader taxonomic comparison background may help.
Despite the mention of parts of the anterior limbs, they were not described, which is relevant because some of the most interesting features of Iniidae are in that region.
My main concern with the manuscript regards the phylogenetic analysis. The size and complexity of the data matrix prevent detailed revision for the journal review time, however, it has several problems that the authors tried to "fix" using a molecular constrained tree, that, unfortunately, does not help much for clarifying the position of the new species here described.

Experimental design

No comment

Validity of the findings

The new species seems to have inioid affinities, but more extensive comparisons should be made.
The phylogenetic analysis is problematic. The matrix used is too large to allow a proper review for the time the journal needs but after running it, it was clear that it has serious problems of character coding. It seems the authors are aware of that because they try to "fix" that with a molecular constrained tree. Despite this may help to resolve the major arrangement of the tree, it certainly does not help for the interrelationships of the smaller groups of the tree, especially when most of the group is extinct, as is the case of the Inioidea.
The use of problematic matrices may explain, for example, why every single new reputed iniid taxa described in the last years comes out as the sister group of Inia, despite some well know species, as Ischyrorhynchus vanbenedeni is by far morphologically close to Inia than any of them.
The use of a large number of ordered characters needs, at least, a detailed justification. None is said about this in the manuscript.
The searching technics used here are quite old and more accurate a modern search algorithms are now available and should be used, including the consideration of use implied weighting since this may help to deal with the large number of homoplasies that a so large matrix obviously contains.

Additional comments

I suggest improving the description including other inioids (ie pontoporids) and basal Delphinida.
The phylogenetic analysis must be revised, perhaps reducing the matrix to a subset of the one used here to better understand the relationships inside Delphinida and reduce the distortions introduced by a lot of homoplasies.
I suggest also to eliminate character state ordination or a careful justification for them.
Using better search algorithms may also help.
Despite I cannot comment in detail, after a rapid analysis of the matrix I am convinced that it is necessary to review codification of several characters.

---

## Round 0.2 · Minor Revisions

Dear Authors,

I have received two reviews, and was waiting for a third one. The last reviewer did not submit the review, but indicated that was happy with your revisions.
I am happy as well, but will recommend minor revision so that you can implement the few points suggested by Dr. Pyenson.
Additionally, there is no specimen loan document attached, so it is not clear if the study material will be returned to Universidade Agostinho Neto or to an Angolan institution. If you can state that explicitly in your response, I would appreciate it.
Otherwise, great to see this study, and I look forward to your revision.
Sincerely,

Tomas Hrbek

·

Basic reporting

I reviewed the initial submission of this manuscript, and I think the revised version is much improved. The authors have done a commendable job at addressing not just my comments and suggestions, but also those of the other two reviewers. Describing important new taxa can be an annoying ordeal when faced with non-overlapping edits and suggestions from different reviewers; the authors have done an excellent job in their Rebuttal document to clearly address all of the concerns. That document, these tracked comments, and most importantly the resultant publication will be a very useful to specialists in their future work — in other words, this work will get cited.

I think the authors have done a satisfactory job including Isthminia in the phylogenetic analyses, explained the elimination of other taxa from the analyses, and included (and discussed) support values for divergences and topologies in their revised manuscript. I think the discussion of comparative phylogenetic results relative to previous studies make the revised manuscript especially useful.

The revised text, figures and tables are solid and really do not require much more attention for publication — see very minor items below.

Experimental design

Again, the authors have mollified my concerns.

One note, on L. 100, the material from Angola is currently on loan to ISEM in France. Neither of the field study approval documents (please note that they use the original taxonomic name from the initial submission) provide any kind of substantive declarations about the status of this material, especially with respect to future disposition in Angola. Will it ever be returned? Under what conditions? I understand the precarious status of natural history institutions in Angola, and I laud the inclusion of many collaborators as co-authors, to increase the stakeholder base, as a means of science diplomacy. But I would say that the authors are ethically obliged to clarify the status of the material not just currently, but their future intents — especially with their open gesture to Angolan culture with their decision for the current species epithet of the new taxon.

Validity of the findings

I think the revised version does a good job fairly parsing the significance of a marine inioid from Angola, relative to all other known inioids. Yes, it would be great if we knew more about the freshwater fossils from South America, but the authors are correct in pointing out that until someone honestly tackles a thorough revision/redescription of putative ischyrorhynchines, we won’t really be able to decipher the timing of marine-to-freshwater transitions in this group. (I say this as someone who has observed and help recode some of this material in Argentine institutions). So, in other words, the revised text is good to go.

Additional comments

Minor comments, but still should be addressed:

L. 208, use indigenous instead of authochtone. I also really encourage the authors to make sure they have contacted members of this group (if they survive) or discussed this species epithet with Angolan citizens. It is an important courtesy.

L. 317, eliminate the stand-alone sentence. It doesn't help the paragraph. Connect it with the preceding one, please.

L. 594, Pan-Inia requires italicizing. I know, it's annoying, but PhyloCode will be happen. One day.

·

Basic reporting

The modifications introduced improved the manuscript.
I have just two observations but are more a matter of opinion and choice.
1. Despite it is true that Ischyrorhynchus has no ear bones preserved (despite one was referred to it) the available specimens in La Plata and Buenos Aires museums provide more information than the specimen here described, which also has no preserved ear bones. So, this is not a good argument to explain the "problematic" position of Ischyrorhynchus in the proposed phylogeny.
2. I insist that I consider a problem to use an extremely large and taxon extensive matrix as the used here. There are in it lots of non-applicable and missing characters which can introduce more "noise" than rather help in the analysis.
But, as I said, this is not an obstacle for the authors to present their position.

Experimental design

None

Validity of the findings

Acceptable

Additional comments

None

---

## Round 0.3 · accepted · Accept

Dear Authors,

Thank you for your revisions. After reading your revised manuscript and your rebuttal letter explaining how you have dealt with the referee comments, I am pleased to inform you that I find the manuscript acceptable for publication in the present stage.

Congratulations on a well done study that makes a significant contribution of our understanding of the evolution Inioid dolphins and how and when South American freshwater habitats were colonized.

Sincerely,

Tomas Hrbek

#